# Arginine methylation catalyzed by PRMT1 is required for B cell activation and differentiation

Simona Infantino[1,2,3], Amanda Light[1,2], Kristy O'Donnell[1,2,3], Vanessa Bryant[1,2], Danielle T. Avery[4], Michael Elliott[5,6], Stuart G. Tangye [4,7], Gabrielle Belz [1,2], Fabienne Mackay[8], Stephane Richard[9] & David Tarlinton[1,2,3]

Arginine methylation catalyzed by protein arginine methyltransferases (PRMT) is a common post-translational modification in mammalian cells, regulating many important functions including cell signalling, proliferation and differentiation. Here we show the role of PRMT1 in B-cell activation and differentiation. PRMT1 expression and activity in human and mouse peripheral B cells increases in response to in vitro or in vivo activation. Deletion of the *Prmt1* gene in mature B cells establishes that although the frequency and phenotype of peripheral B cell subsets seem unaffected, immune responses to T-cell-dependent and -independent antigens are substantially reduced. In vitro activation of *Prmt1*-deficient B cells with a variety of mitogens results in diminished proliferation, differentiation and survival, effects that are correlated with altered signal transduction from the B cell receptor. Thus PRMT1 activity in B cells is required for correct execution of multiple processes that in turn are necessary for humoral immunity.

[1] Walter and Eliza Hall Institute of Medical Research, Parkville, Victoria 3052, Australia. [2] Department of Medical Biology, University of Melbourne, Parkville, Victoria 3010, Australia. [3] Department of Immunology and Pathology, Monash University, Melbourne, Victoria 3004, Australia. [4] Immunology Division, Garvan Institute of Medical Research, Darlinghurst, NSW 2010, Australia. [5] Sydney Medical School, University of Sydney, Sydney, NSW 2006, Australia. [6] Chris O'Brien Lifehouse Cancer Centre, Royal Prince Alfred Hospital, Sydney, NSW 2050, Australia. [7] St Vincent's Clinical School, Faculty of Medicine, University of NSW, Darlinghurst, NSW 2010, Australia. [8] Department of Microbiology and Immunology, University of Melbourne, Parkville, Victoria 3010, Australia. [9] Lady Davis Institute for Medical Research, McGill University, 3755 Cote Ste-Catherine Road, Montreal, Quebec, Canada H3T 1E2. Correspondence and requests for materials should be addressed to S.I. (email: simona.infantino@monash.edu) or to D.T. (email: david.tarlinton@monash.edu)

Methylation of arginine is the most abundant type of protein methylation in mammalian cells and is a major modulator of protein function[1]. The three identified types of modified arginine are monomethylated arginine (MMA), asymmetric dimethylated arginine (ADMA) and symmetric dimethylated arginine (SDMA), all catalyzed by one of the nine known protein arginine methyltransferases (PRMT)[1–3]. ADMA formation is catalyzed by type I PRMTs (comprising PRMT1, 2, 3, 4, 6 and 8), SDMA by type II (PRMT5) and MMA by type III (PRMT7)[1]. The importance of the modifications catalyzed by these enzymes is indicated by embryonic lethality of PRMT1-deficient and PRMT5-deficient mice[4, 5] and by the severe phenotype of PRMT2, PRMT3, PRMT4 and PRMT6-deficient mice[6–9].

PRMT1 is the major arginine methyltransferase active in mammalian cells and is required for normal embryogenesis, cell cycle progression, cell viability, and signal transduction[1, 2, 6, 10]. PRMT1 methylates histones, RNA-binding proteins (RBPs), cell cycle proteins and proteins involved in regulating gene transcription, including high-mobility group proteins (HMGA) and runt-related transcription factor 1 (RUNX1)[1, 2]. PRMT1 is also an active component of signal transduction pathways including those from the B-cell receptor (BCR), T-cell receptor (TCR), nerve growth factor receptor, type 1 interferon receptor, and the pathway involving NF-AT[11–16]. PRMT1 is also known to directly regulate the activity of forkhead box protein O1 (FOXO1)[18] and Bcl-2-associated death promoter (BAD)[17], proteins associated with cell viability. In these examples, the motif RxRxxS/T recognized by protein kinase B (PKB) and present in both FOXO1 and BAD, overlaps with sites recognized by PRMT1, leading to competition in which methylation by PRMT1 inhibits phosphorylation by PKB, thereby prolonging the localization of FOXO1 in the nucleus in one case and inhibiting the pro-apoptotic activity of BAD in the other[17, 18]. Clearly, arginine methylation catalyzed by PRMT1 is crucial to a multitude of pathways although the regulation of its activity is unclear in the majority of cases.

We described previously a PRMT1 target motif within the Ig-α signalling component of the BCR[14]. This evolutionarily conserved arginine is proximal to the immunoreceptor tyrosine activation motif (ITAM) of Ig-α, is methylated directly by PRMT1 in immature B cells, and regulates the activation and differentiation of these cells following BCR ligation[14]. Interestingly, arginine methylation of Ig-α in immature B cells is transiently diminished following BCR ligation, indicating this methylation could be a controlled event and therefore, arginine methylation could contribute to the regulation of B-cell activation in physiological settings[14]. To investigate the possible involvement of PRMT1 in B-cell differentiation further, we undertook an analysis of immune function in mice with Prmt1 deleted specifically in mature B cells. This analysis revealed the post-translational modification of proteins catalysed by PRMT1 to be a regulated event, and to be essential for normal proliferation, differentiation and survival of activated B cells in vitro and for normal antibody responses to antigen in vivo.

## Results

**PRMT1 is dispensable in mature B cells**. We analyzed the function of PRMT1 in mature B cells by generating mice in which Prmt1 was deleted only in the periphery, which also circumvented the congenital lethality of its deficiency[4, 19]. This was done by crossing Prmt1^{f/f} mice with mice in which Cre recombinase was expressed under the control of CD23 regulatory elements[20], thereby initiating deletion at the T2 stage of B-cell development to produce a peripheral B-cell compartment that was Prmt1

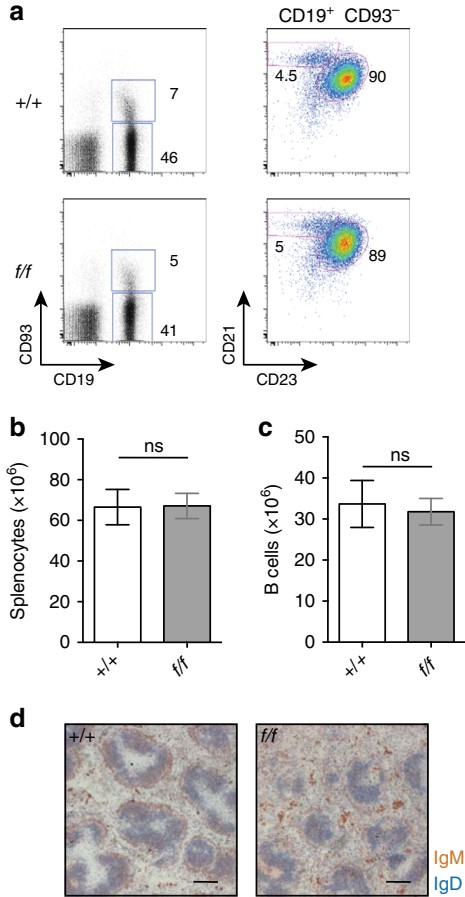

**Fig. 1** Intact CD23[+] B-cell compartment despite deletion of *Prmt1*. **a** Representative flow cytometry plots examining the frequency of mature (CD19[+]CD93[−]), immature (CD19[+]CD93[+]), follicular (CD19[+]CD93[−]CD23[+]CD21[+]) and MZ (CD19[+]CD93[−]CD23[−]CD21[+]) B cells in the spleen of control (+/+) and *Prmt1*^{f/f}CD23Cre (*f/f*) mice. Numbers indicate percentage of the displayed events. **b, c** Number of splenocytes and B cells are indicated. **d** Representative immunohistochemical staining of frozen sections from spleens of control (+/+) and *Prmt1*^{f/f}CD23Cre (*f/f*) naive mice with scale bars indicating 100 μm. All data are representative of three independent experiments and n = 10 mice. Statistical significances were determined using Student's *t*-test; ns = not significant. Mean and s.e.m. in **b** and **c**. Flow cytometry gating strategies for this figure are shown in Supplementary Fig. 6

deficient. Analysis of B-cell development in the spleens of these *Prmt1*^{f/f}CD23Cre mice revealed no abnormalities in B-cell number, phenotype or distribution (Fig. 1). This was true for both immature and mature B cells, distinguished by CD93 expression, and for marginal zone and follicular B cells, resolved by CD21 and CD23 expression (Fig. 1a–c). The localization of B cells in the splenic white pulp also was unaffected by loss of PRMT1 (Fig. 1d). Thus, despite the absolute requirement for PRMT1 in embryogenesis[4], it was not required for the appearance or maintenance of B-cell subsets in the periphery.

**B-cell activation increases PRMT activity**. We next examined the amount of PRMT1 in B cells, both resting and after activation. B cells were purified from the spleens of control and *Prmt1*^{f/f} CD23Cre mice and the amount of PRMT1 assessed by western blot before and after stimulation with CD40L in the presence of interleukins (IL) 4 and 5. PRMT1 was detected in unstimulated control B cells and in increased amounts following activation

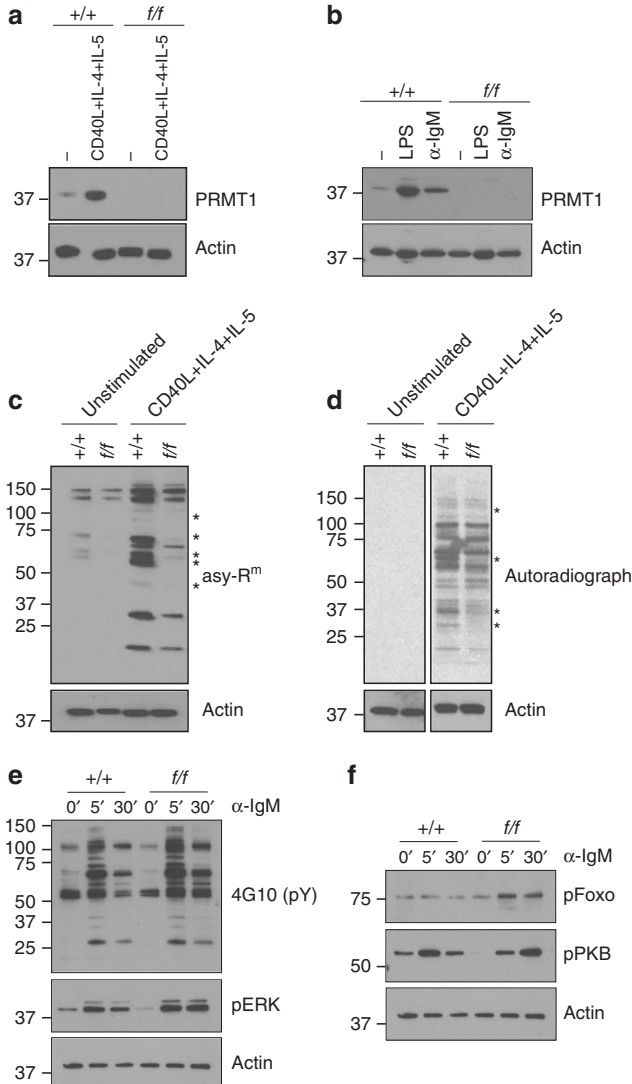

**Fig. 2** Increased PRMT1 and arginine methylated proteins in activated B cells. Immunoblot analysis of control (+/+) and *Prmt1*-deficient (*f/f*) splenic B cells for PRMT1 induction after **a** 2 days stimulation with CD40L and IL-4 and IL-5 or **b** 2 days stimulation with LPS or F(ab′)$_2$ anti-IgM as indicated and compared to actin loading control. **c** Distribution of asymmetric-dimethylated arginine (asy-R$^m$) containing proteins in unstimulated and 2 day CD40L + IL-4 + IL-5 stimulated control (+/+) and *Prmt1*-deficient (*f/f*) splenic B cells, using specific antibody and compared to actin loading control. **d** Ex vivo methylation assay. Unstimulated (left panel) or day 2 CD40L + IL-4 + IL-5 activated (right panel) control (+/+) and *Prmt1*-deficient (*f/f*) B cells were grown for 3 h in the presence of L-[methyl-3H]methionine and protein synthesis inhibitors. Autoradiography of lysates reveals methylated proteins. Asterisks in **c**, **d** indicate proteins that are differentially methylated between control and *Prmt1*-deficient B cells. **e**, **f** B cells were activated with F(ab′)$_2$ anti-IgM for the indicated time points. Western blot analysis of the whole-cell lysates shows the level of phospho-tyrosine (4G10), pERK (T202/Y204), **e** pFoxo1(pS256) and pPKB (S473) **f**. Actin is used as a loading control. Similar results were obtained in three independent experiments in which each sample was derived from a pool of three mice. Uncropped images of blots are presented in Supplementary Fig. 5

(Fig. 2a). Amounts of PRMT1 increased after stimulating control B cells with either lipopolysaccharide (LPS) or F(ab′)$_2$ anti-IgM, albeit to a greater extent with LPS (Fig. 2b). As expected, PRMT1 was not detected in *Prmt1$^{f/f}$ CD23Cre* B cells (Fig. 2a, b). The presence of PRMT1 in control B cells and its increase following activation suggested that PRMT1 activity, and thus the distribution of proteins containing asymmetrically dimethylated arginine, would also change following B-cell stimulation. We assessed PRMT1 activity in resting and activated B cells by two methods. First, total cell lysates from resting and activated, control and *Prmt1$^{f/f}$CD23Cre* B cells were separated by gel electrophoresis and probed for the presence of proteins containing asymmetric dimethylated arginines using a specific antibody. In resting, control B cells, several bands were revealed, indicating constitutive arginine methylation of a subset of proteins (Fig. 2c). Despite the absence of PRMT1, asymmetrically dimethylated proteins were detected in lysate from unstimulated *Prmt1$^{f/f}$ CD23Cre* B cells, but at a frequency and intensity that was less than in control B-cell samples (Fig. 2c), and presumably reflected the activity of other type I PRMTs in these cells. The intensity of asymmetric dimethylated arginine-containing protein bands increased in control B cells following activation with CD40L, coincident with the increased amounts of PRMT1 (Fig. 2a, c). Some bands corresponded in molecular weight to those present in the unstimulated control B-cell sample, but the intensity was increased and new bands were visible (Fig. 2c). The number and intensity of arginine methylated protein bands also increased in CD40L-stimulated *Prmt1*-deficient B cells relative to their unstimulated sample, but again their frequency and intensity were less than in equivalently treated control samples (Fig. 2c). Importantly, unique bands were detected in the stimulated control samples that were not present in the equivalent *Prmt1*-deficient sample, indicating unique PRMT1 substrates in B cells. The second approach assessed ongoing or active methylation by culturing B cells in the presence of both radioactive L-[methyl-3H] methionine and protein synthesis inhibitors. This assay did not detect ongoing methylation in unstimulated B cells from either *Prmt1$^{f/f}$ CD23Cre* or control animals, but the activity increased significantly in both genotypes following stimulation (Fig. 2d). Again, the extent and intensity of labelling differed between control and *Prmt1*-deficient B cells, reflecting increased methyltransferase activity in the control B-cell sample. Collectively these results indicate that PRMT1 is a major asymmetric dimethyltransferase in B cells, that its activity increases following stimulation and that it has unique substrates in B cells.

In light of the role PRMT1 activity has in regulating signalling events downstream of the BCR in immature B cells[14], we examined the consequences of PRMT1 deficiency on signal transduction following BCR ligation on mature, naive B cells. Control and *Prmt1*-deficient B cells were stimulated with F(ab′)$_2$ anti-IgM, then lysed after 0, 5 or 30 min and the distribution of tyrosine phosphorylated proteins determined (Fig. 2e). At time zero, the few tyrosine phosphorylated protein bands were distributed similarly in both B-cell samples. After 5 min stimulation, the intensity and number of such bands had increased substantially in both *Prmt1*-deficient and control B-cell samples, again with similar distributions. At 30 min post stimulation, the intensity of bands in both samples was reduced from their 5 min peaks, however the reduction was less in *Prmt1*-deficient B cells than in controls, resulting in a relative hyper-phosphorylation in the absence of PRMT1 (Fig. 2e). Interestingly we also found increased phosphorylation of ERK1/2, FOXO1 and PKB in equivalently anti-IgM stimulated *Prmt1*-deficient B cells (Fig. 2e, f). These results revealed activation-induced increases in PRMT1 activity under a variety of conditions and an apparent prolongation of signalling from the

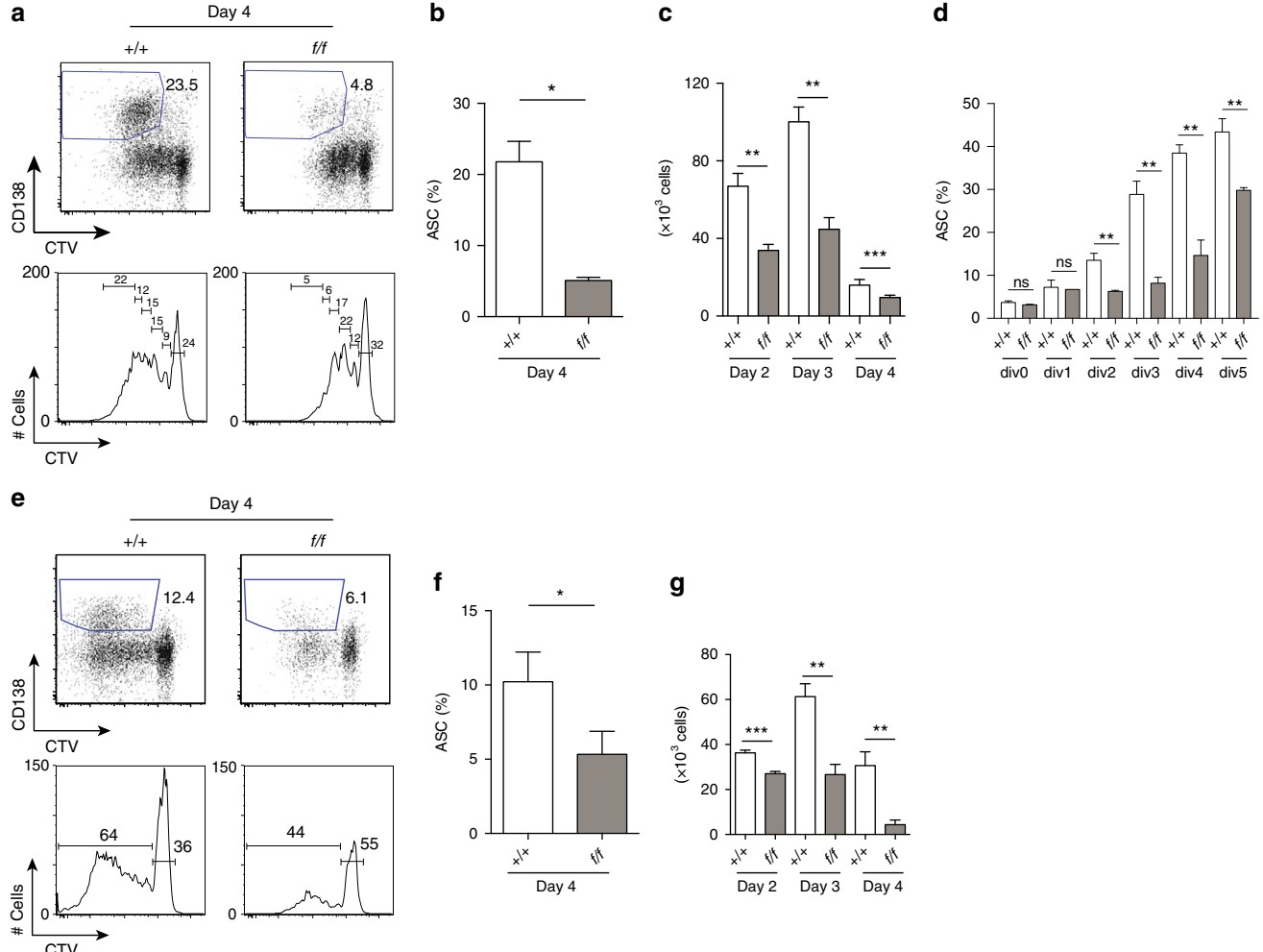

**Fig. 3** Defective response of *Prmt1*-deficient B cells to in vitro stimulation. Control (+/+) and *Prmt1^{f/f}CD23Cre* (f/f) splenic B cells, labelled with CTV, were activated with **a–d** CD40L + IL-4 + IL-5 or **e–g** LPS (25 µg ml^{−1}) for the indicated times. **a, e** Cultured cells were monitored for ASC differentiation with anti-CD138 antibody (top panels) and extent of cell division with CTV dilution (bottom panels). Percentages are the proportion of viable cells within indicated regions. **b, f** Frequency of live, CD138^+ ASC at 4 days after activation. **c, g** Number of viable cells in cultures after stimulation for the indicated times. **d** Frequency of CD138^+ cells within the indicated cell divisions of CD40L + IL-4 + IL-5 stimulated control (+/+) and *Prmt1^{f/f}CD23Cre* (f/f) splenic B cells at day 4. $*P \leq 0.05$ $**P \leq 0.01$, $***P \leq 0.001$ (unpaired *t*-test). One representative experiment of three, with three biological replicates, is shown. (Mean and s.e.m in **b–d**, **f**, **g**). Flow cytometry gating strategies for this figure are shown in Supplementary Fig. 7

BCR in the absence of PRMT1. We also examined CD19^+ B cells isolated from human peripheral blood for PRMT1 amounts before and after CD40L + IL-4 activation and found the amount also increased following stimulation (Supplementary Fig. 1a).

**PRMT1 deficiency affects B-cell proliferation and differentiation.** The increased amount of PRMT1 in B cells after stimulation suggested a possible role for protein arginine methylation in B-cell activation. To assess this, we labelled control and *Prmt1*-deficient mouse B cells with the division tracking dye, CTV, and then cultured the cells with CD40L, IL-4 and IL-5. The extent of B-cell proliferation and differentiation were compared after 4 days using, respectively, the distribution of cell divisions and the frequency of antibody secreting cell (ASC) as indicated by CD138 expression[21]. This analysis revealed reduced proliferation and differentiation in the *Prmt1^{f/f} CD23Cre* B-cell cultures (Fig. 3a, b). The impact of *Prmt1* deficiency on proliferation was apparent throughout the culture, as assessed by counting the number of B cells on successive days (Fig. 3c). To separate effects on

proliferation from differentiation, which are intimately linked[22], we assessed the division profiles of control and *Prmt1*-deficient B cells after 4 days stimulation with CD40L, IL-4 and IL-5 for the relative frequency of CD138^+ cells, thereby normalizing for proliferation (Fig. 3d). While this revealed division-dependent increases in differentiation in both control and *Prmt1*-deficient B-cell cultures, there was significantly less differentiation per division in the latter. Equivalent experiments using LPS stimulation (Fig. 3e–g) also revealed significantly less proliferation and differentiation in *Prmt1*-deficient B cells compared to controls, which was apparent again over the course of the culture (Fig. 3e–g). Thus *Prmt1* deficiency affected both B-cell proliferation and differentiation, although the effect on the former appeared to be more marked.

Lymphocyte activation induces metabolic changes that have profound effects on cell proliferation and differentiation[23]. In B cells, for example, metabolic reprogramming is required for antibody production[24]. Given the requirement for PRMT1 in normal B-cell activation, next we compared the metabolic pathways utilized in control and *Prmt1*-deficient B cells. This

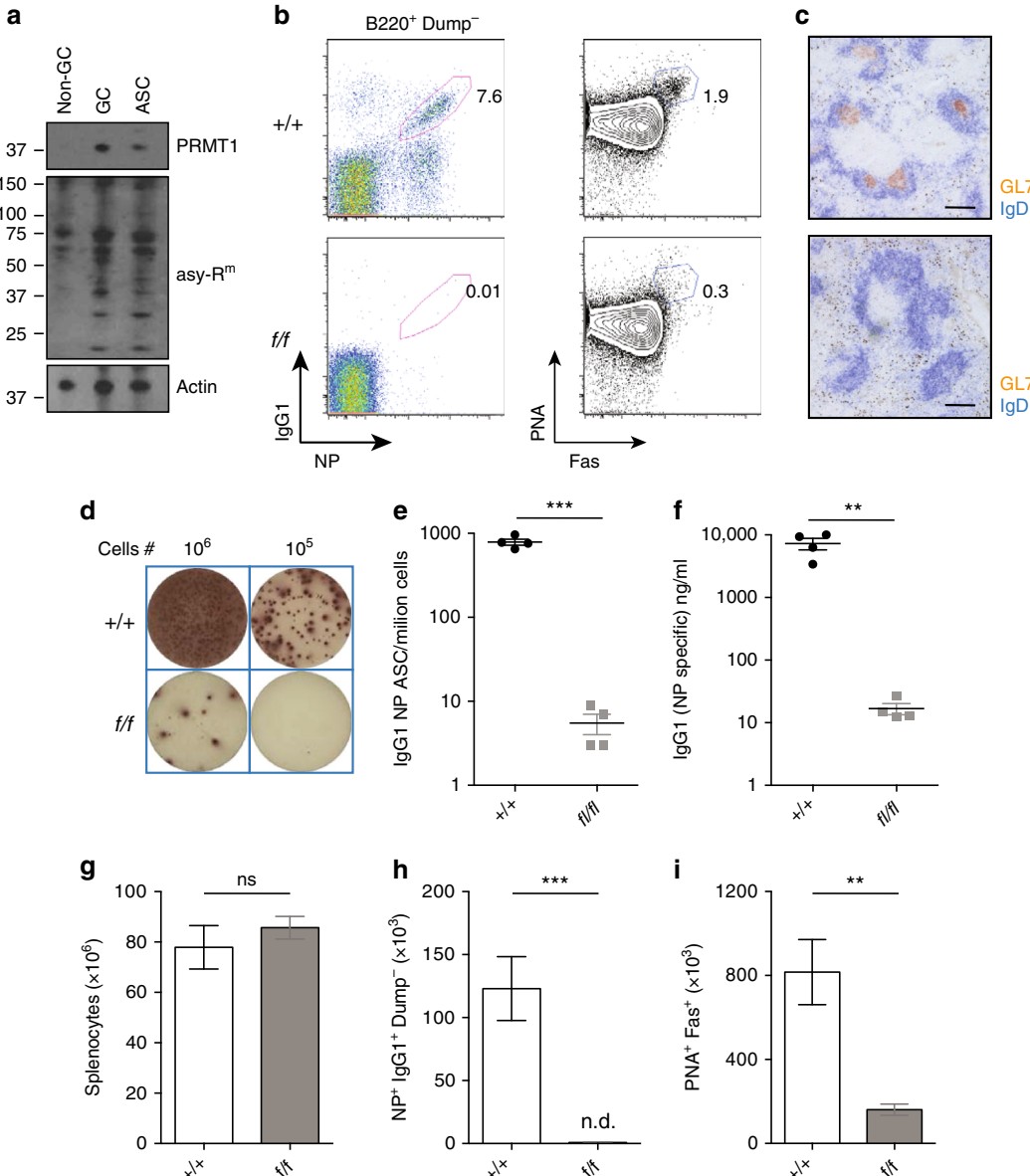

**Fig. 4** PRMT1 is required for GC formation in response to immunization. **a** Control mice were immunized with NP-KLH in alum and the indicated B-cell subsets were sorted from spleen 7 days post immunization, lysed and proteins separated by SDS-PAGE. Western blot analysis shows PRMT1 (top panel), asymmetrically methylated arginine-containing proteins (middle panel) and actin (loading control, bottom panel) in non-GC B cells (CD19$^+$Fas$^-$PNA$^-$), GC B cells (CD19$^+$Fas$^+$PNA$^+$) and ASC (CD138$^+$B220$^{low}$). **b-i** Control (+/+) and Prmt1$^{f/f}$CD23Cre (f/f) mice were immunized with NP-KLH in alum and analyzed at day 7. **b** NP-specific IgG1$^+$ B cells (left) and total GC B cells (right) are shown in representative flow cytometry plots. Numbers indicate frequency of indicated events within the B220$^+$Dump$^-$. Dump antibodies detected IgM; IgD; Gr1. **c** Immunohistochemical staining for GC in spleen sections at day 7 post NP-KLH immunization using GL7 (orange) and IgD (blue) with scale bars indicating 100 μm. Representative wells **d** and frequency per million splenocytes **e** of NP-specific IgG1$^+$ ASC as measured by ELISpot at day 7 post immunization. **f** Serum titres of NP-specific IgG1 antibody as determined by ELISA at day 7. Total splenocyte number **g**, number of NP-reactive IgG1$^+$ B cells **h** and GC B cells (PNA$^+$Fas$^+$) **i**, calculated at day 7 post immunization of control (+/+) and Prmt1$^{f/f}$CD23Cre (f/f) mice as indicated. **P ≤ 0.01, ***P ≤ 0.001; ns = not significant (unpaired t-test). Data are representative of three independent experiments with four mice per group. Mean and s.e.m. in **e-i**. Uncropped images of blots are presented in Supplementary Fig. 5. Flow cytometry gating strategies for this figure are shown in Supplementary Fig. 8

revealed that the basal oxygen-consumption rate (OCR) in unstimulated B cells was not influenced by Prmt1 deficiency (Supplementary Fig. 2a) but conversely, PRMT1 was required for basal and maximal respiratory capacity as well as glycolytic capacity in activated B cells (Supplementary Fig. 2a–c). Thus, PRMT1 activity was required for normal B-cell responses to stimuli that mimic aspects of T-cell dependent (TD) or T-cell independent (TI) responses and this included the metabolic reprogramming that follows activation.

**PRMT1 is required in B cells for humoral immunity**. The increased amounts of PRMT1 and asymmetrically arginine methylated proteins following in vitro activation of B cells suggested that similar changes might occur in in vivo activated B cells. We purified therefore naive B cells, germinal centre (GC) B cells and ASC from mice 7 days after immunization with a TD antigen, prepared lysates and probed each sample for both PRMT1 and proteins containing asymmetrically dimethylated arginines (Fig. 4a). The amount of PRMT1 was low in non-GC

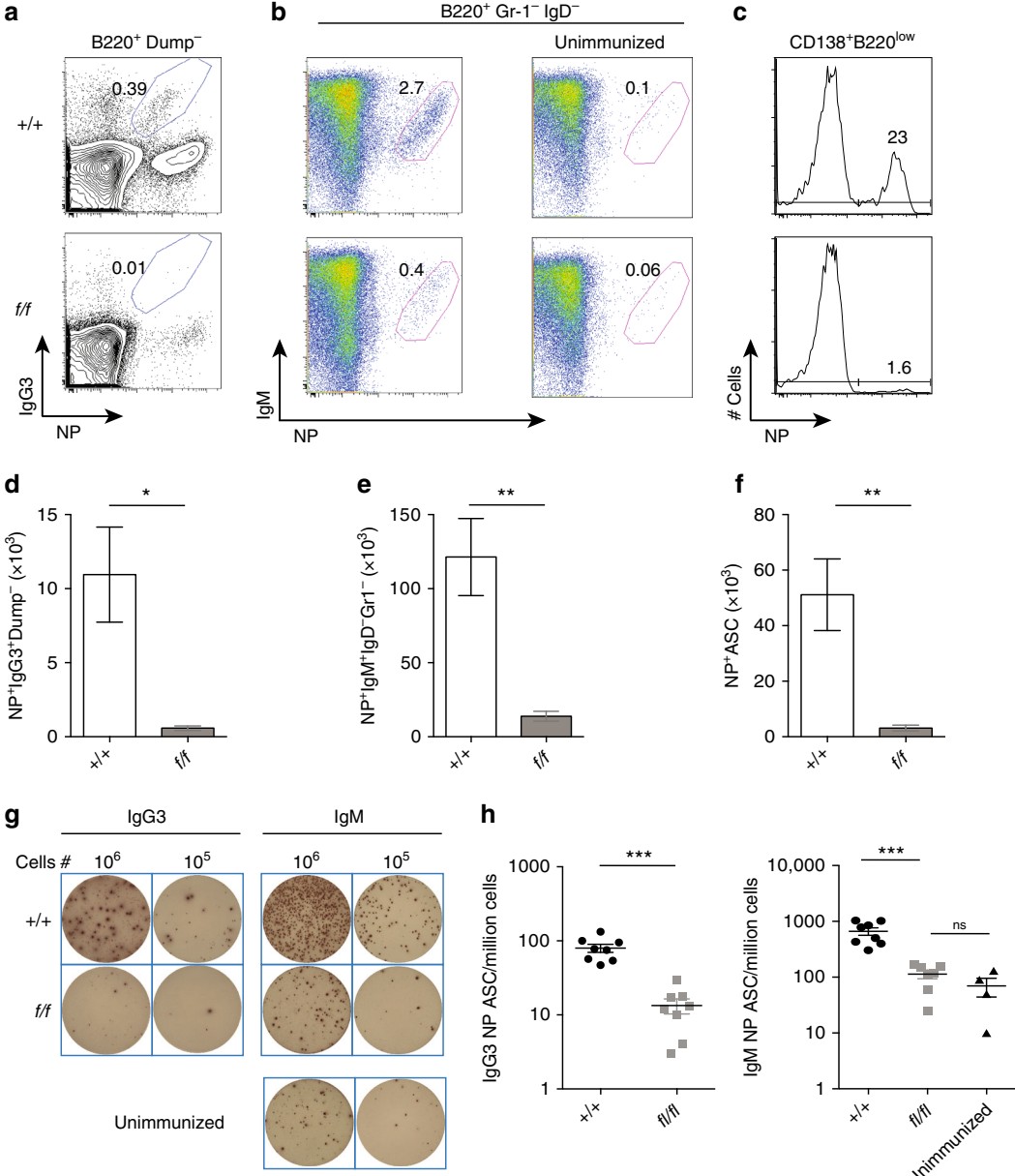

**Fig. 5** PRMT1 is critical for a TI immune response. Control (+/+) and *Prmt1*$^{f/f}$*CD23Cre* (*f/f*) mice were injected i.p. with NP-Ficoll and analysed after 5 days. Splenic NP-specific **a** IgG3+ and **b** IgM+ B cells are shown in representative flow cytometry plots along with an unimmunized mouse for IgM in (upper) control (+/+) and (lower) *Prmt1*$^{f/f}$*CD23Cre* (*f/f*) mice. **c** Histograms display the proportion of spleen ASC (CD138+B220$^{low}$) that were NP-specific at day 5 post immunization in (upper) control (+/+) or (lower) *Prmt1*$^{f/f}$*CD23Cre* (*f/f*) mice. **a**–**c** Numbers represent the frequency of events within the indicated regions. **d**–**f** Numbers of splenic NP-specific IgG3+ B cells **d**, NP-specific IgM+ B cells **e** and NP-specific ASC **f** are shown. Representative wells **g** and frequency **h** of NP-specific IgG3+ and IgM+ ASC as measured by ELISpot in the spleen and presented as ASC per million spleen cells. Naive IgM ELISpot sample **g** is from control (+/+) mice. *$P \leq 0.05$; **$P \leq 0.01$; ***$P \leq 0.001$; ns = not significant (unpaired *t*-test). Data are representative of two independent experiments with four mice per group. Mean and s.e.m. in **d**–**f** and **h**. Flow cytometry gating strategies for this figure are shown in Supplementary Fig. 9

splenic B cells, but increased in both GC B cells and, to a lesser extent, in ASC. Probing lysates for asymmetrically arginine-methylated proteins revealed few bands in naive B cells, but these increased in number and amount in both the GC B cell and ASC samples (Fig. 4a). We purified also naive (CD20+CD27−CD38$^{lo}$), GC (CD20$^{hi}$CD27+CD38$^{hi}$) and memory (CD20+CD27+CD38$^{lo}$) B cells from human tonsils and found PRMT1 to be readily detected in all samples, although the amount was slightly increased in GC B cells (Supplementary Fig. 1b). Examining these samples for proteins containing asymmetric dimethlyated

arginines revealed a similar pattern in all cell types, but with some bands of increased intensity in the memory and GC B-cell fractions (Supplementary Fig. 1c). Thus, human B cells ex vivo also showed activation-dependent increases in PRMT1 amounts and protein arginine methylation.

To assess possible functions of PRMT1 in B-cell responses in vivo, *Prmt1*$^{f/f}$ *CD23Cre* and control mice were immunized intraperitoneally with the TD antigen (4-hydroxy-3-nitrophenyl) acetyl coupled to keyhole limpet hemocyanin (NP-KLH) delivered with an alum adjuvant. In the ensuing immune

response, we monitored the frequency of isotype-switched, antigen-specific B cells and total GC B cells (Fig. 4b). Remarkably, NP⁺IgG1⁺ B cells, PNA⁺FAS⁺ GC B cells and physical GC structures were not detected in the spleens of immunized *Prmt1^{f/f}CD23Cre* mice, in contrast to their abundance in control mice (Fig. 4b, c). Whether *Prmt1*-deficient B cells generated NP-specific IgG1⁺ ASC in spleen or NP-specific serum antibody in vivo was determined by ELISpot assays and ELISA, which revealed a more than 100-fold reduction in both ASC frequency and antibody titres in *Prmt1^{f/f} CD23Cre* mice (Fig. 4d–f). Importantly, these differences were reflected as absolute numbers (Fig. 4g–i) with similar differences recorded 5 and 14 days after immunization. To determine whether a defective immune response was restricted to hapten antigens, we infected *Prmt1^{f/f} CD23Cre* mice with influenza virus and found similar defects in GC and ASC production (Supplementary Fig. 3). In this case, both *Prmt1^{f/f} CD23Cre* and control mice increased the number of CD8⁺ D^bNP366-binding cells, confirming successful infection and an absence of any global immune suppression in *Prmt1^{f/f}CD23Cre* mice (Supplementary Fig. 3d, e).

*Prmt1^{f/f} CD23Cre* mice were also challenged with NP-Ficoll, a TI antigen, and again failed to generate a normal response (Fig. 5). While the frequency of NP⁺IgM⁺ B cells increased in the spleens of *Prmt1^{f/f} CD23Cre* mice following this immunization (Fig. 5b), indicating a degree of clonal expansion, this was substantially less than occurred in controls, as was the amount of NP-specific IgG3 antibody in serum (Fig. 5d–h). Thus, responses to both TD and TI antigens were defective in mice with *Prmt1*-deficient B cells.

**Immune defects are strictly B-cell intrinsic.** The immune defects identified in mice with *Prmt1*-deficient B cells could in principle arise from an inability of such B cells to induce appropriate CD4⁺ T-cell help, potentially compounding an otherwise modest B-cell defect. We examined this possibility by creating BM radiation chimeras with 50% of hematopoietic cells derived from Ly5.2 *Prmt1^{f/f} CD23Cre* BM and 50% from Ly5.1 congenic BM that was *Prmt1* wild type. Control chimeras were made with equal amounts of Ly5.2 *CD23Cre* and Ly5.1 BM, all cells being *Prmt1* sufficient. Nine weeks after reconstitution, both groups of mice were immunized with NP-KLH in alum and analyzed 7 days later for the presence of antigen-specific B cells of either allotype in the spleen (Supplementary Fig 4). *Prmt1*-deficient B cells, although equally represented in the naïve B-cell population, failed to generate detectable NP⁺IgG1⁺ B cells, despite the presence of control NP⁺IgG1⁺ B cells and T-cell help (Supplementary Fig. 4a, b). The same *Prmt1*-deficient B-cell deficit was apparent in the NP-specific ASC compartment, identified as NP⁺CD138⁺B220^{lo} and then partitioned by expression of Ly5.1 (Supplementary Fig. 4c, d). In control chimeras, both allotypes were represented equally in the NP-reactive B-cell and ASC populations (Supplementary Fig. 4). The immune response defect was still apparent when we examined all GC B cells, irrespective of specificity. Thus, *Prmt1*-deficient B cells were unable to either initiate or enter into a GC reaction.

**PRMT1 is required in B cells for a memory response.** The previous experiments revealed a requirement for PRMT1 in B cells for normal humoral immune responses. These data, however, did not test whether PRMT1 was required in memory B cells to mount a recall response. To assess this requirement, we first created mice in which *Prmt1* deficiency was both inducible and restricted to the B-cell lineage. This was done by reconstituting lethally irradiated recipients with a BM mixture comprising 80% from mice homozygous for the μMT mutation, and thus unable to generate B cells[25], and 20% from mice that were *Prmt1^{f/f}* and

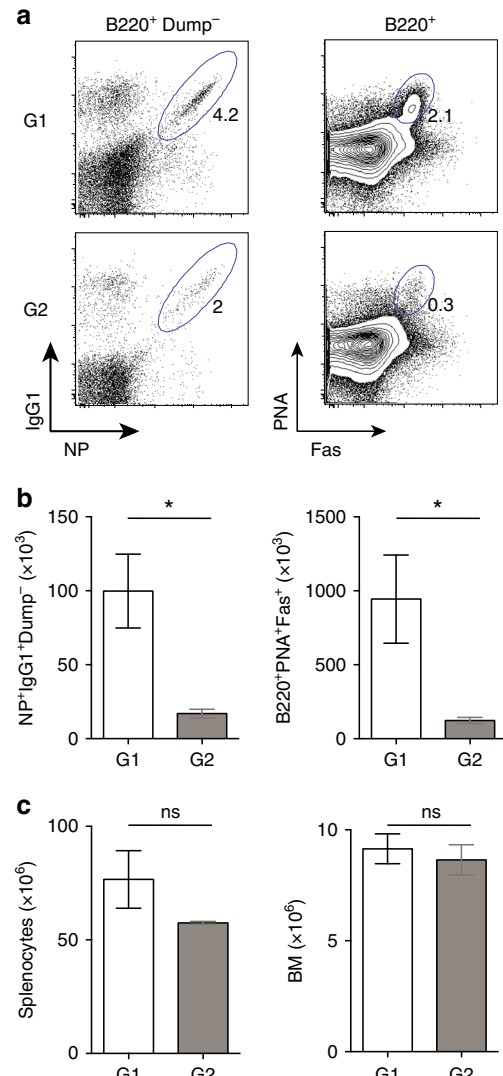

**Fig. 6** PRMT1 is required to mount a recall immune response. **a–c** Irradiation chimeras were generated with 80% of μMT and either 20% of *Prmt1^{+/+}Rosa26CreERT2* (G1;+/+) or 20% *Prmt1^{f/f}Rosa26CreERT2* (G2; f/f) BM cells. Nine weeks post BM reconstitution; mice were immunized with NP-KLH in alum. Nine weeks after immunization mice were treated for 3 days with single doses of Tamoxifen to induce *Prmt1* deletion and then boosted with NP-KLH in PBS, injected i.p. **a** Representative flow cytometry plots of splenocytes 5 days after NP-KLH boost are shown. Frequencies of NP⁺IgG1⁺ (left) and total GC (B220⁺Fas⁺PNA⁺) (right) among B220⁺IgM⁻IgD⁻Gr1⁻ cells and B220⁺ cells, respectively, are indicated. Numbers represent percentages of the displayed events within indicated gates. **b**, **c** Average numbers of NP⁺IgG1⁺ B-cells, GC B cells, total splenocytes per spleen and per femur are shown in the graphs. *P ≤ 0.05; ns = not significant (unpaired *t*-test). Mean and s.e.m. (**b**, **c**). The experiment comprised five mice per group. Flow cytometry gating strategies for this figure are shown in Supplementary Fig. 10

carried the *Rosa26CreERT2* allele encoding a constitutively expressed, Tamoxifen inducible form of Cre[26]. In control chimeric mice, the 20% BM donor was *Prmt1^{+/+} Rosa26CreERT2*. After reconstitution, these two groups of mice were immunized with NP-KLH in alum and 9 weeks later, dosed on 3 successive days with Tamoxifen to induce activation of Cre and deletion of the *Prmt1^{f/f}* alleles. The mice were then boosted with soluble NP-KLH and examined 5 days later for the presence of both NP-specific B cells and GC B cells (Fig. 6). Boosted mice in which

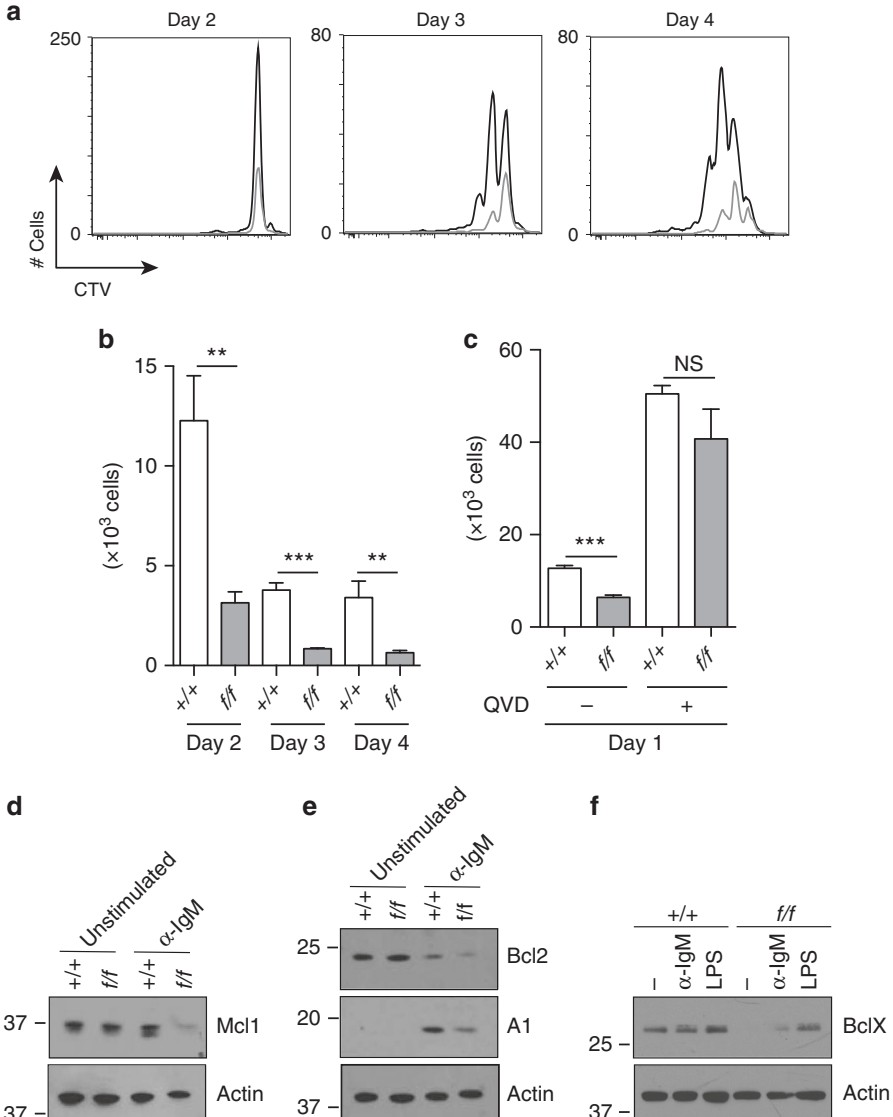

**Fig. 7** *Prmt1*-deficient B cells respond abnormally to in vitro stimulation. **a**, **b** Splenic B cells, purified from control (+/+) and *Prmt1*[f/f]*CD23Cre* (f/f) mice, were labelled with CTV and cultured in the presence of F(ab')₂ anti-IgM antibody for the indicated times. **a** Overlay histograms show division profiles of live cells at days 2, 3 and 4 of activation for control (black line) and *Prmt1*-deficient (grey line) B cells. **b** Quantification of live cells at time points indicated after stimulation for control (+/+) and *Prmt1*-deficient (f/f) samples. **c** Control (+/+) and *Prmt1*-deficient B cells (f/f) were activated with F(ab')₂ anti-IgM antibody for 24 h in the presence or absence of the broad caspase inhibitor, QVD. Viable cell numbers are indicated. **b**, **c** \*\*$P \leq 0.01$; \*\*\*$P \leq 0.001$; ns = not significant (unpaired *t*-test). Data are representative of three experiments with three replicates per experiment. Mean and s.e.m. **d–f** Defective regulation of MCL1, BCL2, A1 and BCLX in activated *Prmt1*-deficient B cells. Immunoblot analysis of spleen B cells from control (+/+) and *Prmt1*[f/f]*CD23Cre* (f/f) mice. **d**, **e** B cells were unstimulated or activated for 24 h with F(ab')₂ anti-IgM, then lysed and separated by SDS-PAGE. **d** MCL1 and **e** BCL2 and A1 protein amounts were detected with the respective antibodies. **f** Control (+/+) and *Prmt1*[f/f]*CD23Cre* (f/f) spleen B cells were unstimulated (−), activated with F(ab')₂ anti-IgM (α-IgM) or LPS for 24 h, then lysed and separated by SDS-PAGE. BCLX protein was detected by specific antibody as indicated. **d–f** Actin was used as a loading control and comparable results were obtained in three independent experiments. Uncropped images of blots are presented in Supplementary Fig. 5. Flow cytometry gating strategies for this figure are shown in Supplementary Fig. 11

*Prmt1* had been deleted from memory B cells showed a significant deficit in NP⁺IgG1⁺ and GC B cells compared to controls in which *Prmt1* was retained (Fig. 6a, b). Importantly, this treatment did not significantly reduce the overall number of B cells in spleen or BM (Fig. 6c). Thus, as in a primary immune response, PRMT1 activity was required for a normal memory B-cell response.

## PRMT1 regulates survival following BCR ligation.
B-cell responses to TD antigens depend on stimulation through CD40L and the BCR. While the reduced proliferation and differentiation of *Prmt1*-deficient B cells in response to stimulation with CD40L and cytokines could contribute to defective immunity, it was possible that other pathways were affected. To determine whether *Prmt1* deficiency altered proliferation following BCR ligation, B cells purified from spleens of *Prmt1*[f/f]*CD23Cre* and control mice were labelled with CTV and cultured for 4 days in the presence of F(ab')₂ anti-IgM to crosslink the BCR (Fig. 7a, b). On days 2, 3 and 4, the number of cells present in the cultures and the extent of cell division were assessed, revealing a reduced number of divisions, a reduced number of cells in each division and significantly reduced overall viability at

all times for *Prmt1*-deficient B cells compared to controls (Fig. 7a, b). The nature of the cell death induced by BCR ligation was examined by repeating the cultures in the presence of the pan-caspase inhibitor, QVD, which reduced cell death at 24 h such that viability in the *Prmt1^{f/f} CD23Cre* and control cultures was equal (Fig. 7c).

That QVD blocked the increased apoptosis in *Prmt1*-deficient B cells following BCR stimulation suggested a connection between PRMT1, BCR ligation and the proteins that protect against activation induced cell death[27]. We examined therefore the amounts of BCL2 family members MCL1, BCL2, A1 and BCLX in *Prmt1^{f/f} CD23Cre* and control B cells before and after stimulation with anti-IgM (Fig. 7d–f). MCL1, which was equally abundant in unstimulated *Prmt1^{f/f}CD23Cre* and control B cells, was reduced in *Prmt1*-deficient B cells following BCR ligation. BCL2 decreased in both control and *Prmt1^{f/f} CD23Cre* B cells following BCR ligation, but to a greater extent in *Prmt1*-deficient B cells. A1 was induced in both groups, but less so in *Prmt1^{f/f} CD23Cre* B cells (Fig. 7e). BCLX was detectable in equal amounts prior to and after both stimulations in control B cells, but was barely detectable in unstimulated *Prmt1*-deficient B cells and increased marginally following BCR ligation (Fig. 7f). Thus, PRMT1 was required for appropriate regulation of expression of pro-survival members of the BCL2 family in B cells before and after BCR stimulation such that in its absence, apoptosis was significantly increased.

## Discussion

PRMT1, considered the predominant type I arginine methyl-transferase activity in mammalian cells, regulates a multitude of molecular and cellular functions[2]. This includes gene transcription, responding to DNA damage, mRNA translation, cell division, apoptosis and signal transduction[14, 18, 19, 28, 29]. Despite such individual examples, potential roles for PRMT1 in B-cell immune responses have not previously been reported. In the present study we have examined the role of PRMT1 in mature B cells. We have shown that the increased activity of PRMT1 upon B-cell activation serves to co-ordinate a number of crucial processes that are required for a successful immune response and that are dysregulated in its absence. These dysregulated processes include cell cycle progression, metabolism and survival, the last two being novel findings of this work. Indeed, the altered expression of BCL2 family pro-survival proteins in the absence of PRMT1 could be a major contributor to the immunodeficiency we have observed in these mice, as MCL1, one of the altered proteins, is essential for GC B-cell and plasma cell survival[26, 30].

Asymmetrically dimethylated arginine proteins are present in naive B cells before increasing in amount significantly following B-cell activation. Given that we found protein methylation activity to be essentially undetectable in naive B cells over a 3-h interval in vitro, this result implies that arginine methylation is a particularly stable post-translational change in these cells. Interestingly, we observed several novel asymmetrically methylated protein bands in both control and *Prmt1*-deficient B-cell samples following stimulation, indicating unique and redundant substrates of PRMT1 that presumably participate in the processes that are dysregulated in the enzyme's absence. Identifying these substrates and the activities that are dependent on PRMT1 will be the basis of significant future work, determining how methylation modulates their roles in proliferation, differentiation and survival following B-cell activation by a variety of stimuli.

Activation of *Prmt1*-deficient B cells is associated also with significant apoptosis. This cell death could reflect the defect in mitochondrial respiratory capacity, meaning that PRMT1 is required for either the metabolic changes that sustain B-cell

growth and viability following activation, or to integrate those changes into a viability signal. The marked dysregulation in expression of pro-survival members of the BCL2 family of proteins observed in response to BCR stimulation of *Prmt1*-deficient B cells could also contribute to increased apoptosis. Specifically, the inability of *Prmt1*-deficient B cells to sustain MCL1 and BCL2 proteins following BCR ligation, or to upregulate and/or maintain A1 protein and BCLX could all contribute to increased cell death after BCR ligation. Indeed, the ability of the broad caspase inhibitor, QVD, to block apoptosis of *Prmt1*-deficient B cells is consistent with such a role. These results reveal a previously undescribed role for PRMT1 in cell death. Collectively, impairments in proliferation, differentiation, metabolism and survival could combine to create the near-complete block in B-cell immune responses observed in vivo.

The consequences on in vivo B-cell behaviour of *Prmt1* deletion are not restricted by the nature of the antigen. We observed a deficit of plasma cells and GC B cells in mice containing *Prmt1*-deficient B cells following either immunization with a protein antigen in adjuvant or infection with influenza virus. In response to immunization, antigen-specific, isotype-switched antibody is detected in the serum of *Prmt1*-deficient mice along with rare IgG1 ASC in spleen, suggesting at a minimum the initiation of an immune response. That we measured equal expansion of influenza specific CD8^+ T cells in control and *Prmt1^{f/f}CD23Cre* mice following infection rules out technical issues or systemic immune suppression in the *Prmt1*-deficient group. Similarly, observing a degree of clonal expansion in response to NP-Ficoll, indicates that antigen specific *Prmt1*-deficient B cells were exposed to and at least transiently activated by antigen. Reflecting these in vivo results, in vitro proliferation of *Prmt1*-deficient B-cells in response to a range of mitogens, although significantly less than in controls, does occur to some extent. We also investigated the requirement for PRMT1 in a memory B-cell response to a TD antigen and found it again to be indispensable. Our finding that PRMT1 protein expression and activity in human B cells mirrored that in mice B cells—being higher in GC and plasma cells compared to naive B cells and being induced by in vitro activation —suggests the roles of PRMT1 in B cells will be conserved between these species and that inhibiting PRMT1 activity may be beneficial in situations of pathogenic B-cell hyperactivity.

Arginine methylation within the cytoplasmic tail of Ig-α alters the consequences of ligation of the pre-BCR, increasing the activity of PI3K and thus promoting differentiation over proliferation[14]. We showed here in mature B cells that BCR ligation in the absence of PRMT1 also leads to elevated and prolonged PI3K activation, as indicated by PKB phosphorylation. Similarly, and perhaps counterintuitively given the overall hypo-responsiveness at the cell level, total protein tyrosine phosphorylation was increased and prolonged in PRMT1-deficient B cells, as was phosphorylation of FOXO1 and ERK1/2. These results suggest that in the period shortly after BCR stimulation, PRMT1 acts as a negative regulator of signalling in a role that is apparent in both pre-B and follicular B cells.

PRMT1 regulates the activity and/or persistence of FOXO1 by controlling the access of PKB[17, 18]. Interestingly, FOXO1 is active in GC B cells where its predominant role is to establish the dark zone (DZ)[28, 31]. FOXO1 is required also for class switch recombination and affinity maturation in vivo[28, 31]. Equally, PI3K activity is an important regulator of both GC formation and class switch recombination, promoting cell survival and proliferation on one hand and repressing AID activity on another[32]. The compromised appearance of GC and class switched, antigen-specific B-cells in mice carrying *Prmt1*-deficient B cells could be explained in part by reduced FOXO1 arginine methylation leading to increased phosphorylation by PKB and thus

inactivation. These consequences of not methylating FOXO1 could be amplified by the extended PI3K activity we observed in *Prmt1*-deficient B cells following BCR ligation. Thus, the combined effects of dysregulated FOXO1 and PI3K activity could perturb B-cell proliferation, class switch recombination, plasma cell differentiation and ultimately the GC reaction. Determining the precise mechanisms by which PRMT1 activity regulates different components of B-cell activation will be a challenging but essential part of understanding how this enzyme coordinates multiple aspects of B-cell biology.

PRMT1 has many substrates[2], which may explain why its absence has such profound, heterogeneous and often cell type- and stage-specific effects. For example, PRMT1 could be involved in regulating proliferation through its action on MRE11, which is thought to contribute to the chromosomal instability and cell cycle checkpoint abnormalities seen in PRMT1 deficient embryonic fibroblasts[19, 33]. More recently, PRMT1 was shown to methylate cyclin-dependent kinase 4 (CDK4) in FOXO1-expressing pre-B cells, preventing or disrupting complexes between CDK4 and cyclin-D3, and thereby blocking cell cycle progression and promoting differentiation[34]. It is possible that either or both these activities contribute to the deficit of B-cell expansion observed here in vitro and in vivo and further work will identify the relevance of these mechanisms in mature B cells.

It is possible also that the consequences of *Prmt1* deletion in B cells are mediated not by the loss of PRMT1 activity per se, but rather through a concomitant alteration in the activity of another PRMT, which is what actually affects the GC reaction and plasma cell differentiation[35]. The activity of the transcription factor E2F-1, for example, varies depending on the relative amounts of PRMT1 and PRMT5, which compete for access to the same methylation site, promoting apoptosis and proliferation, respectively[36]. In addition to the role of PRMT1 in GC formation reported here, PRMT7 has been shown to repress transcription of *Bcl-6* via H4R3 methylation in its promoter[37]. Deletion of *Prmt7* in B cells increases *Bcl6* expression, promoting GC development and repressing *Irf4* and *Prdm1*, genes that are required for plasma cells differentiation[37]. Conceivably, any interplay between PRMT1 and PRMT7 in which one affected the activity of the other, could contribute to the in vivo B-cell phenotype observed here. The intersection of the activities of the different PRMT, which may be substrate specific, suggests an assessment of all remaining arginine methylation activity may be required to fully understand the mechanisms underlying the phenotypes observed in the absence of any individual PRMT.

A previous report has described normal TD responses and abnormal TI responses in B-cell *Prmt1*-deficient mice[38]. B cells from these mice were also found to be normal or even augmented in their proliferation in response to mitogen stimulation in vitro. However, these earlier studies differed from those reported here in two respects. First, *Prmt1* was deleted in the earliest stages of B-cell development using a CD19-Cre, potentially enabling compensation of its absence during development. Second, *Prmt1* deletion was incomplete with PRMT1 protein detected in peripheral B cells, which means some of the reported results may be dosage related[38].

In conclusion, our results identify PRMT1 as a central regulator of humoral immunity, establishing a previously unidentified role of this enzyme in the activation, proliferation and differentiation of B cells. Our findings have demonstrated the dynamic induction of arginine methylation in activated B cells and have revealed striking similarities in PRMT1 expression and activity between mouse and human B cells, suggesting conserved roles in supporting proper immune responses in both species. We have also established that, in activated B cells, PRMT1 is required to sustain MCL1 and BCL2 protein expression, revealing a

previously unknown role of PRMT1 in apoptosis. Overall, our studies indicate that PRMT1 activity may be essential to integrate multiple pathways into a coherent outcome that is manifest as both B cell activation and an immune response.

## Methods

**Mice and immunizations.** *Cd23Cre* mice[20] were provided by Meinrad Busslinger and the creation of *Prmt1^{fl/fl}* mice has been described[19]. All mice were maintained at the Walter and Eliza Hall Institute of Medical Research (WEHI) on a C57BL/6 background. Animal procedures were approved by the WEHI Animal Ethics Committee and mice used in the study were male or female, and 8 weeks of age on commencement. No animals were specifically excluded from the study and the investigators were not blinded to the groups during the experiments or analyses. For primary immune responses, mice were injected intraperitoneally (i.p.) with 100 µg of 4-hydroxy-3-nitrophenyl)acetyl (NP) conjugated to keyhole limpet hemocyanin (KLH) at a molar ratio of (13-20):1, precipitated on 10% alum. For secondary responses, mice were boosted i.p. with 50 µg of NP-KLH in PBS. Sample sizes were calculated to detect with 80% power a difference of at least 30% in the means with a variance of 20%, with equal numbers in all groups. For influenza infections mice were inoculated with $10^4$ p.f.u. of HKx31 (H3N2) influenza virus[39].

**Flow cytometry, antibodies and cell purification.** Single cells were resuspended in PBS 2% FCS and stained for flow cytometric analysis, with dilution of antibodies ranging between 1:100 and 1:800. The following reagents were used: Ly5.2 (104), CD138 (281), IgG1 (X56), Fas/CD95 (Jo2), Ly5.1 (A20.1) and CD19 (1D3) from Becton Dickinson Biosciences (BD); PNA from Vector Laboratories; IgG3 from Southern Biotech; CD93 (AA4.1) from eBioscience; GL7 (GL7), Gr1 (8C5), IgM (331.12), IgD (11/26), CD21 (76G), CD23 (B3B4), CD8 (53-6.7) and B220 (RA3-6B2) were conjugated in-house. Human antibodies: CD20 (2H7), CD27 (M-T271) and CD38 (HIT2) from BD. Virus-specific CD8$^+$ T cells were detected with tetrameric H-2b major histocompatibility complex with the influenza virus nucleoprotein peptide (NP; H-2Db-restricted NP 336-374)[39]. FcγRII/III (24G2; supernatant) was used to block non-specific antibody binding. For western blot and in vitro activation, cells were analyzed live (with the addition of propidium iodide) on the FACS CantoII (Becton Dickinson-BD) and data analyzed using Flowjo software (Treestar). Splenic B cells from control (+/+) or (f/f) mice were obtained using the B-cell isolation kit and LS magnetic columns, following the manufacturer's protocol (Miltenyi Biotech GmbH). Human B cells were isolated from PBMC or tonsils using CD19 microbeads and LS magnetic columns (Miltenyi Biotech) and further processed as indicated. B-cell purity (>98%) was determined using CD19 and B220 antibodies. Sort-purification: cells were stained with antibodies as indicated and purified by FACS Aria or Influx (BD), with purity >98%. Mouse naive (non-GC) (CD19$^+$Fas$^-$PNA$^-$), GC (CD19$^+$Fas$^+$PNA$^+$) and plasma cells (CD138$^+$B220$^{low}$) were sorted from spleen 7 days after NP-KLH immunization. Naive B cells (CD20$^+$CD27$^-$CD38$^{lo}$), memory B cells (CD20$^+$CD27$^+$CD38$^{lo}$) and GC B cells (CD20$^+$CD38$^{hi}$CD27$^+$) were sorted to >98% purity from human tonsils using the indicated antibodies. Tonsillar tissues were collected following informed consent from patients undergoing routine tonsillectomy (Mater Hospital, North Sydney, Australia). Approval for this study was obtained from the human research ethics committees of the St. Vincent's Hospital and Sydney South West Area Health Service, NSW, Australia.

**Bone marrow chimeras.** To generate 50:50 chimeras, lethally irradiated Ly5.1 mice (2 × 5.5 Gy) were reconstituted with a mixture of 50% Ly5.1 BM and 50% *Prmt1^{fl/fl}*CD23$^{Cre/+}$ or CD23$^{Cre/+}$ BM. Mice were rested for 7–8 weeks before NP-KLH/alum immunization as described above. For µMT chimeras, lethally irradiated Ly5.1 mice (2 × 5.5 Gy) were reconstituted with 80% µMT BM and 20% *Prmt1^{fl/fl}*ERT2$^{Cre/+}$ or ERT2$^{Cre/+}$ BM. Mice were rested for 9 weeks and then immunized with NP-KLH in alum. Nine weeks after immunization, mice were treated three times (1 dose per day at days 1, 2 and 8) with Tamoxifen to induce activation of Cre[40], and then at day 10 boosted with NP-KLH in PBS, injected i.p. Mice were bled before immunization to test chimerism by flow cytometry.

**Histology, ELISPOT and ELISA.** Portions of spleen were frozen in OCT (Tissue-Tek Sakura), and 7 µm sections were cut with a microtome (Leica) at −20 °C. Sections were fixed in ice-cold acetone for 10 min, air-dried, then stained as indicated in the figures with goat anti-mouse IgM conjugated to HRP (Southern Biotech) at 1:200 dilution, biotinylated anti-IgD (clone 1126c; Southern Biotech) at 1:100 dilution, or anti-GL7 (clone GL7; made in-house) at 1:200 dilution. Biotinylated antibodies were revealed using streptavidin alkaline phosphatase (Southern Biotech) at 1:200 dilution. Unlabelled antibody was detected with anti-rat kappa HRP (Southern Biotech) at 1:200 dilution. Staining was visualised with an AEC substrate kit (Vector Laboratories) and Vector Blue substrate kit (Vector Laboratories)[41] and slides were mounted using Aqua Polymount (Polysciences, Inc.). ASC and serum antibody titres were analyzed by ELISPOT and ELISA, respectively[41]. For ELISA, at the times indicated blood was collected and serum separated. 96-well plates were coated with 20 µg ml$^{-1}$ NP$_{13}$-BSA (conjugated in-house), and diluted serum was incubated for at least 20 h at room temperature. NP-specific IgG1 was

detected with goat anti-mouse IgG1-HRP (Southern Biotech) at 1:500 dilution and visualized with ABTS substrate (2,2-Azinobis [3-ethylbenzthiazoline Sulfonic Acid]; Sigma-Aldrich). For ELISpots, splenocytes were added to a 96-well cellulose ester membrane plate (Millipore) coated with 20 $\mu$g ml$^{-1}$ of NP$_{13}$-BSA and incubated for up to 20 h at 37 °C and 5–10% CO$_2$. Anti-NP-specific IgG$_1$, IgM or IgG$_3$ were revealed using goat anti-mouse antibodies conjugated to HRP at 1:500 dilution (all from Southern Biotech), and visualized with substrate AEC (3-amino-9-ethyl carbazole; Sigma-Aldrich). Spots were counted using an automated reader (AID ELISpot Reader System, software version 4).

**Cell lysates and western blot**. Mouse and human B cells were lysed on ice for 30 min in Ex-250 lysis buffer (20 mM Hepes, pH 7.5, 250 mM NaCl, 0.5 mM MgCl$_2$, 0.5% NP-40, complete protease inhibitors (Roche) and 500 $\mu$M sodium orthovanadate). Lysates were centrifuged for 10 min at 1000×$g$, and the supernatant was further diluted with Ex-00 (20 mM Hepes, pH 7.5, 0.5 mM MgCl$_2$, 0.5% NP-40) to obtain a final concentration of 150 mM NaCl. This was centrifuged for 20 min at 20,000×$g$. Human tonsil naive (CD20$^+$CD27$^-$CD38$^{lo}$), memory (CD20$^+$CD27$^+$CD38$^{lo}$) and GC (CD20$^+$CD38$^{hi}$CD27$^+$) B cells were lysed on ice for 30 min in lysis buffer containing protease/phosphatase inhibitors (1% NP-40 in 10 mM Tris-HCL, 150 mM NaCl, 0.1% NaN$_3$, pH 7.8) and centrifuged for 20 min at 14,000 rpm. Lysates were separated by SDS-PAGE and transferred to membranes that were probed with the following antibodies: BCL2 (3F11; BD Biosciences), MCL1 (19C4-15; WEHI), A1 (6D6; WEHI)[42], Actin (goat polyclonal; Santa Cruz), PRMT1 (rabbit polyclonal; Cell Signalling), phospho-Foxo1 (Ser256, rabbit polyclonal; Cell Signalling), phospho-PKB (Ser473, rabbit clone 193H12; Cell Signalling), phospho-p44/42 (pERK1/2, Thr202/Tyr204, rabbit polyclonal; Cell Signalling), BCLX (44/Bcl-x; mouse monoclonal; BD); dimethyl-arginine antibody (ASYM24; rabbit polyclonal; Millipore), phospho-tyrosine (4G10; Millipore). Western blots were developed using an in-house enhanced chemiluminescence system.

**B-cell activation**. B cells were activated with the following stimuli: CD40L (1:1000 membrane preparation from Sf21 cells[22]) in the presence of recombinant mouse IL-4 (10 ng ml$^{-1}$; R&D Systems) and IL-5 (5 ng ml$^{-1}$; R&D Systems); F(ab′)$_2$ anti-IgM antibody (20 $\mu$g ml$^{-1}$; Jackson ImmunoResearch); or LPS (25 $\mu$g ml$^{-1}$; Difco) for the indicated times. Pancaspase inhibitor Q-VD (MP Biomedicals) was used at a final concentration of 10 $\mu$M. Human B cells were stimulated with 100 ng ml$^{-1}$ of Mega CD40L (Enzo) in the presence of 50 ng ml$^{-1}$ recombinant human IL-4 (Peprotech).

**In vivo methylation assay**. Naive or activated B cells were incubated for 30 min in RPMI 1640 complete medium without methionine (Gibco) containing 20 $\mu$g ml$^{-1}$ chloramphenicol and 100 $\mu$g ml$^{-1}$ cycloheximide (Sigma-Aldrich), after which the cells were labelled with 10 $\mu$Ci ml$^{-1}$ of L-[methyl-3H] methionine[14]. The cells were incubated for an additional 3 h in the presence of the methyl group donor and the same protein synthesis inhibitors. The cells were lysed, equal samples loaded on a gel and blotted onto a nitrocellulose membrane. Methionine incorporation was determined by autoradiography.

**Statistical analyses**. Data sets were compared for significant differences using appropriate tests and as indicated in the relevant figure legends. Variances between groups that were being statistically compared were similar.

**Data availability**. The data that support the findings of this study are available from the corresponding author upon request.

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

## Acknowledgements

We thank members of the Tarlinton lab for critical review of the manuscript, discussion and general technical assistance. This work is supported by a National Health and Medical Research Council (NHMRC) Australia program grant to D.T. and G.B. (1054925). S.I. was supported by a DRFG Fellowship, D.T. by an NHMRC Research Fellowship and G.B. by an Australian Research Council Future Fellowship. This work was made possible through Victorian State Government Operational Infrastructure Support and Australian Government NHMRC IRIISS.

## Author contributions

S.I. and D.T. designed the research; S.I., A.L. and K.O.D. performed experiments and analyzed the data with D.T.; V.B., D.T.A., M.E., S.R., S.G.T., F.M. and G.B. provided reagents, resources and intellectual input; S.I. and D.T. wrote the manuscript, which was edited by all authors.

## Additional information

**Competing interests:** The authors declare no competing financial interests.

