## [Peer review file · Nature Communications]

Reviewers' comments:

Reviewer #1 (Remarks to the Author):

General

The role of arginine methylation in immunomodulation is an important yet understudied facet of immunology. Here, Infantino et al show that in mature B cells loss of PRMT1 does not affect B cell homeostasis, consistent with their finding that it is only modestly expressed in resting B cells but induced upon activation (with anti-CD40/IL-4). In the absence of PRMT1, primary and secondary antibody responses and the GC response are decimated, which is a striking finding of some importance. However, it is unclear why this is the case since previously characterized substrates such as Ig-alpha and Foxo1 are not investigated, and new physiologic targets are not identified. The latter may not be an easy task since the substrates are likely numerous, resulting in pleiotropic effects, but some resolution would be helpful. In addition, the study lacks biochemical analyses to support the observed defects in CD40/LPS/BCR signaling and reduced expression of Bcl2/Mcl1. Hence, the study has merit but the manuscript in its current form is underdeveloped and a bit disjointed.

Specific

1. Figure 2 shows variable expression of PRMT1 in naïve and stimulated or GC B cells. It is difficult to discern what is representative and thus quantitation is needed. Moreover, differential arginine methylation in human B cell subsets is not evident.
2. Is enhanced expression of PRMT1 in Fig. 2 unique to CD40 stimulation? Comparisons should be made with anti-IgM and LPS.
3. Providing evidence for PRMT1 activity in B cells is not a very compelling stand alone result. Identification of the methylated targets observed in Fig. 2 would add novelty and mechanistic insight to the work (but perhaps is above and beyond that which is required for publication in N. Communications).
4. Figure 3. Are the effects on plasma cell differentiation in response to LPS/CD40/IL-4 due to proliferation, survival and/or differentiation factors? What is the impact on downstream signaling (PI3K/NF-kB/MAPK pathways)?
5. Using small molecule inhibitors Infantino et al previously showed that Arg methylation of Ig α impaired BCR signaling. Thus PRMT1^{-/-} B cells should be hyperresponsive to BCR signaling. Moreover, arginine methylation of Foxo1 by PRMT1 prevents efficient phosphorylation by Akt, thus in the absence of PRMT1 Foxo should have heightened activity due to its sustained presence in the nucleus. These studies should be performed to provide a context for interpreting the current findings.
6. Panel 4a is out of sequence and should be moved to Fig. 1
7. Fig 5 could be supplemental as it addresses a rather unlikely possibility, especially given that the PRMT1^{-/-} B cells were also defective in T-independent responses.
8. The TI-2 responses (Fig. S3) should be included in the main text.
9. The effects of PRMT1 loss on BCR-induced proliferation and survival are interesting but inexplicable. Does PRMT1 regulate the stability of Bcl2/Mcl1? What is the effect on Bcl-x, which is prominently induced by BCR engagement. What is the effect on downstream signaling via the BCR? As above, previous findings by Infantino et al would suggest that PRMT1 B cells are hyperresponsive to BCR signaling. How is this explained?

Reviewer #2 (Remarks to the Author):

A. Summary of the key results

The manuscript entitled "Arginine methylation catalyzed by PRMT1 is required for B cell activation and differentiation" by Infantino et al. describes that PRMT1 catalyzes arginine methylation in

activated-B cells, where it plays a role in immune responses. The authors demonstrated that distribution and appearance of B cells are not changed by loss of PRMT1 in the periphery. However, levels of PRMT1 expression and activity were increased as the B cells become activated. In addition, the authors also provide interesting information to suggest that PRMT1 as a central molecule regulating humoral immunity is necessary for mature B cells proliferation, differentiation, and persistence following activation in a range of circumstances. Finally, the authors suggested that PRMT1 regulates the survival of B cells via modulation of Bcl2 family proteins expression including Mcl-1 and Bcl-2 as a previously unknown role of PRMT1 in apoptosis.

B. Originality and interest

This study provides a novel link between the regulation of humoral immunity and PRMT1 activity that is really interesting and potentially an excellent addition to cellular roles for protein arginine methylation in multiple fields.

C. Data & methodology

Overall, experimentation technically sounds and is sufficiently valid. Biophysical and biochemical experiments are carefully conducted, yielding clear results.

D. Appropriate use of statistics and treatment of uncertainties

Data are exquisite that are used with statistical analysis, and the interpretations are appropriately cautious.

E. Conclusion: robustness, validity, reliability

This study has been well designed and performed, and provides a very interesting viewpoint in understanding between the role of PRMT1-mediated protein arginine methylation and humoral immunity.

F. Suggested improvements: experiments, data for possible revision

This study provides interesting and potentially important sights into the impact of protein arginine methylation on the regulation of immune functions. However, I have following concerns. In the experiment shown in Figures 2 and 3, the authors suggested that PRMT1 expression and activity are increased following stimulation with CD40L and cytokines, which is required B cell proliferation and differentiation.

1. It would be interesting to know whether phenotypes of Prmt1-deleted B cell are rescued with the expression of wild type PRMT1, but not methyltransferase-inactive mutants.

2. This study lacks mechanistic insight into the role of PRMT1. How can PRMT1 regulate two different cellular events, proliferation and differentiation on the same stimulation? The authors need to investigate and discuss the molecular mechanism such as intracellular signal transduction, target substrates and gene expression. For example, as the authors showed that de novo methylated substrates of PRMT1 with stimulation (Figure 2c), if they identify methylated proteins, it should be helpful for understanding the molecular basis of the mechanism.

G. References: appropriate credit to previous work?

There are already enough References essential for the manuscript.

H. Clarity and context: lucidity of abstract/summary, appropriateness of abstract, introduction and conclusions

Abstract, introduction and conclusions were clearly written and well controlled in the manuscript.

Point by Point Response

Reviewer #1 (Remarks to the Author):

General

The role of arginine methylation in immunomodulation is an important yet understudied facet of immunology. Here, Infantino et al show that in mature B cells loss of PRMT1 does not affect B cell homeostasis, consistent with their finding that it is only modestly expressed in resting B cells but induced upon activation (with anti-CD40/IL-4). In the absence of PRMT1, primary and secondary antibody responses and the GC response are decimated, which is a striking finding of some importance. However, it is unclear why this is the case since previously characterized substrates such as Ig-alpha and Foxo1 are not investigated, and new physiologic targets are not identified. The latter may not be an easy task since the substrates are likely numerous, resulting in pleiotropic effects, but some resolution would be helpful. In addition, the study lacks biochemical analyses to support the observed defects in CD40/LPS/BCR signaling and reduced expression of Bcl2/Mcl1. Hence, the study has merit but the manuscript in its current form is underdeveloped and a bit disjointed.

We thank reviewer 1 for their appreciation of the significance of the work and note the General comments they have made, which could improve the scope and impact of the work.

In regard to our having not examined Foxo1 and Ig α , we have now included an assessment of FOXO1 in anti-IgM stimulated B cells (Fig. 2f). Ig- α , however, remains beyond our ability as the antibody that detects its specific arginine methylation, made in the Reth laboratory in Germany, is no longer available. These changes are on page 8.

With respect to determining new, physiological substrates for PRMT1 in B cells, unfortunately we have not been able to address this in a substantive manner. We are gearing up for proteomics studies of B cells but as we get further into the planning, it is clear that these are whole projects on their own. We will, having identified differentially methylated proteins, determine which arginine is the target, what the consequences are of that methylation on the proteins function or half-life, and then which of the components of the complex phenotype of the knockout is due to that specific alteration. As we hope the referee is aware, pretty much every substrate so characterised, at least to date, is a publication by itself. We did not feel that we could wait that long but fully appreciate the importance of the topic.

In regard to an apparent lack of biochemical analyses we have now added to our previous biochemistry by including:

- a) total tyrosine phosphorylation after BCR ligation (Figure 2e);*
- b) FOXO1, PKB and ERK phosphorylation after BCR ligation (Figure 2e, f)*
- c) measured PRMT1 amounts after IgM and LPS stimulation (Figure 2b)*
- d) measured BCLX amount after BCR stimulation (Figure 7f)*

We feel these additions have added another dimension to the scope of the work and provided both potential explanations for some of the observed phenomena but also provide a link to previous studies and identify significant areas of future investigation. We appreciate the suggestion to do these experiments. These additions are on pages 7-8 and page 14.

Specific

1. Figure 2 shows variable expression of PRMT1 in naïve and stimulated or GC B cells. It

is difficult to discern what is representative and thus quantitation is needed. Moreover, differential arginine methylation in human B cell subsets is not evident.

We have moved the data from ex vivo activated B cells to Figure 4a and reserve Figure 2 now for in vitro stimulated B cells. We think the induction of PRMT1 is clear and unequivocal in these cells as it is human cells stimulated in vitro (Supplementary Figure 1a). The in vivo induction of PRMT1 is also very clear for mouse (Figure 4a) but less so for in vivo human tonsil samples (Supplementary Figure 1b,c). We don't think this is an issue of quantification but more of biology in that in humans, this change in PRMT1 and its activity may be all that occurs. We have moved the human data to Supplementary Figure 1, in keeping with reviewer suggestions, to better maintain the focus and flow of the manuscript.

2. Is enhanced expression of PRMT1 in Fig. 2 unique to CD40 stimulation? Comparisons should be made with anti-IgM and LPS.

We have now added anti-IgM and LPS to CD40L (Figure 2b) and it is clear that PRMT1 induction is universal. Page 6.

3. Providing evidence for PRMT1 activity in B cells is not a very compelling stand alone result. Identification of the methylated targets observed in Fig. 2 would add novelty and mechanistic insight to the work (but perhaps is above and beyond that which is required for publication in N. Communications).

Please see our reply to General Comments above.

4. Figure 3. Are the effects on plasma cell differentiation in response to LPS/CD40/IL-4 due to proliferation, survival and/or differentiation factors?

What is the impact on downstream signaling (PI3K/NF- κ B/MAPK pathways)?

Good point. We have reanalysed these experiments (and repeated them), using the division profiles revealed by CTV to measure B cell differentiation on a per division basis (Figure 3d). Doing this normalises for proliferation and, in this case, reveals that the defect in differentiation has both a division based component and an intrinsic differentiation component. The sum of these two effects may well be the almost complete block in differentiation we observe in the intact animals. These experiments are described on pages 8-9. The impact of arginine methylation deficiency on PI3K and ERK is now described in Figure 2e,f. NF κ B was unaffected.

5. Using small molecule inhibitors Infantino et al previously showed that Arg methylation of Ig α impaired BCR signaling. Thus PRMT1^{-/-} B cells should be hyperresponsive to BCR signaling. Moreover, arginine methylation of Foxo1 by PRMT1 prevents efficient phosphorylation by Akt, thus in the absence of PRMT1 Foxo should have heightened activity due to its sustained presence in the nucleus. These studies should be performed to provide a context for interpreting the current findings.

These turned out to be remarkably prescient predictions. We have now found evidence of hyper-responsiveness to BCR ligation by extended tyrosine phosphorylation and PKB phosphorylation (Figure 2). We have also found, as predicted, extended phosphorylation of FOXO1 in BCR stimulated PRMT1-deficient B cells. We think this provides a very elegant connection to the previous pre-B cell work, and potentially provides insight into the mechanism by which PRMT1 acts in B cells at least sometimes. It may not, however, explain the totality of the phenotype as this mouse looks unlike all other hyper-responsive animals. Determining the basis of this global phenotype is a significant part of our current effort. The Foxo1 data are described on page 8.

6. Panel 4a is out of sequence and should be moved to Fig. 1

We have left panel 4a in Figure 4 as it fits much better with the remaining in vivo characterisation that starts with that figure. Figure 1 is concerned only with the knockout mice at rest. We do feel,

however, that consolidating the human studies into the one figure, now Supplementary Fig. 4, has greatly improved the flow of the manuscript.

7. Fig 5 (TI immunization) could be supplemental as it addresses a rather unlikely possibility, especially given that the PRMT1^{-/-} B cells were also defective in T-independent responses.

8. The TI-2 responses (Fig. S3) should be included in the main text.

We have moved the TI response to the main text (new Figure 5), moved the 50:50 chimera immune response data to Supplementary Figure 4 as suggested. We have also moved the human data to Supplementary Figure 1, allowing it to be viewed together and not interrupting the mouse narrative with excursions into humans. We remain of the view, however, that this is an important comparison to make and think it worthwhile keeping in the report. This extends over several pages.

9. The effects of PRMT1 loss on BCR-induced proliferation and survival are interesting but inexplicable. Does PRMT1 regulate the stability of Bcl2/Mcl1? What is the effect on Bcl-x, which is prominently induced by BCR engagement. What is the effect on downstream signaling via the BCR? As above, previous findings by Infantino et al would suggest that PRMT1 B cells are hyperresponsive to BCR signaling. How is this explained?

We have added BCR-induced induction of BCLX in control and Prmt1-deficient B cells (Figure 7f).

While we did not see the strong induction of BCLX in control B cells with anti-IgM over unstimulated cells expected by the reviewer, we did see induction with LPS. Prmt1-deficient B cells, however, had little BCLX in unstimulated B-cells, which was only weakly induced by anti-IgM stimulation and somewhat more so by LPS (Figure 7f). The hyper-responsiveness of the B cells is we think now apparent, at least at the signalling level, but this does not translate into global hyper-responsiveness of all genes, gene products or cell processes. We don't know exactly why, but does fit with the proposal we put forward that PRMT1 methylation is a global integrator of signalling in activated lymphocytes, allowing multiple processes to occur at one but be focussed on the one outcome – promoting lymphocyte proliferation and differentiation. The study of BCLX is reported on page 14.

Reviewer #2 (Remarks to the Author):

A. Summary of the key results

The manuscript entitled “Arginine methylation catalyzed by PRMT1 is required for B cell activation and differentiation” by Infantino et al. describes that PRMT1 catalyzes arginine methylation in activated-B cells, where it plays a role in immune responses. The authors demonstrated that distribution and appearance of B cells are not changed by loss of PRMT1 in the periphery. However, levels of PRMT1 expression and activity were increased as the B cells become activated. In addition, the authors also provide interesting information to suggest that PRMT1 as a central molecule regulating humoral immunity is necessary for mature B cells proliferation, differentiation, and persistence following activation in a range of circumstances. Finally, the authors suggested that PRMT1 regulates the survival of B cells via modulation of Bcl2 family proteins expression including Mcl-1 and Bcl-2 as a previously unknown role of PRMT1 in apoptosis.

B. Originality and interest

This study provides a novel link between the regulation of humoral immunity and PRMT1 activity that is really interesting and potentially an excellent addition to cellular roles for protein arginine methylation in multiple fields.

C. Data & methodology

Overall, experimentation technically sounds and is sufficiently valid. Biophysical and biochemical experiments are carefully conducted, yielding clear results.

D. Appropriate use of statistics and treatment of uncertainties

Data are exquisite that are used with statistical analysis, and the interpretations are appropriately cautious.

E. Conclusion: robustness, validity, reliability

This study has been well designed and performed, and provides a very interesting viewpoint in understanding between the role of PRMT1-mediated protein arginine methylation and humoral immunity.

F. Suggested improvements: experiments, data for possible revision

This study provides interesting and potentially important sights into the impact of protein arginine methylation on the regulation of immune functions. However, I have following concerns. In the experiment shown in Figures 2 and 3, the authors suggested that PRMT1 expression and activity are increased following stimulation with CD40L and cytokines, which is required B cell proliferation and differentiation.

1. It would be interesting to know whether phenotypes of Prmt1-deleted B cell are rescued with the expression of wild type PRMT1, but not methyltransferase-inactive mutants. *This is a good suggestion but we have not attempted to do this experiment either in vitro or in vivo. In part, this is because of the difficulty in infecting B cells in general with retroviruses at high efficiency and perhaps especially those that proliferate poorly, such as those described here. There are as yet no PRMT1 transgenic mice of which we are aware and making bone marrow chimeras with retroviruses in stem cells would be problematic, given the probable importance of PRMT1 in multiple hematopoietic lineages.*

2. This study lacks mechanistic insight into the role of PRMT1. How can PRMT1 regulate two different cellular events, proliferation and differentiation on the same stimulation? The authors need to investigate and discuss the molecular mechanism such as intracellular signal transduction, target substrates and gene expression. For example, as the authors showed that de novo methylated substrates of PRMT1 with stimulation (Figure 2c), if they identify methylated proteins, it should be helpful for understanding the molecular basis of the mechanism.

Again, we agree with the sentiment in this suggestion – identifying substrates of PRMT1 in B cells, verifying their nature and the role of PRMT1 methylation in carrying out those functions – would be and will be of great importance. It is, however, simply beyond our capabilities at this point within a reasonable timeframe. We are working on this as we also think this is a really interesting and fruitful area of future research. We would add that reporting our current data will accelerate this endeavour by hopefully promoting other laboratories with expertise in biochemistry and protein methylation to enter into this area.

G. References: appropriate credit to previous work?

There are already enough References essential for the manuscript.

H. Clarity and context: lucidity of abstract/summary, appropriateness of abstract, introduction and conclusions

Abstract, introduction and conclusions were clearly written and well controlled in the manuscript.

We have made a number of minor changes to the manuscript that we think improve the readability and accessibility of the work. These have not altered the data or our interpretation of that data so have not been listed in this point by point. Equally, we have added sections to the discussion to include the recent description of PRMT1 being involved in pre-B cell development by regulating cell cycle, a study in which we collaborated, and we have added a brief discussion on the potential importance of the signalling changes we have now observed. These are on pages 17-18 and 19, respectively. We have edited the original discussion to incorporate these new topics without unduly increasing the length. We don't think any of the salient points have been lost in doing this.

REVIEWERS' COMMENTS:

Reviewer #1 (Remarks to the Author):

The authors did an admirable job in responding to the comments in terms of new data and re-organization of the manuscript. The manuscript is of appropriate scope and content for Nature Communications.

Point by Point Response

Reviewer #1 (Remarks to the Author):

REVIEWERS' COMMENTS:

Reviewer #1 (Remarks to the Author):

The authors did an admirable job in responding to the comments in terms of new data and re-organization of the manuscript. The manuscript is of appropriate scope and content for Nature Communications.

We thank the referee for the response and also for the insightful and helpful comments they provided on the original version.